# Functional Characterization of Two Novel Intron 4 *SERPING1* Gene Splice Site Pathogenic Variants in Families with Hereditary Angioedema

**DOI:** 10.3390/biomedicines12010072

**Published:** 2023-12-28

**Authors:** Olga Shchagina, Elena Gracheva, Alyona Chukhrova, Elena Bliznets, Igor Bychkov, Sergey Kutsev, Aleksander Polyakov

**Affiliations:** 1Research Centre for Medical Genetics, 115522 Moscow, Russiabliznetzelena@mail.ru (E.B.); bychkov.nbo@gmail.com (I.B.); kutsev@mail.ru (S.K.);; 2Department of Health of Vologda Region, Budgetary Healthcare Institution, Vologda Region Regional Clinical Hospital, 160002 Vologda, Russia; grachevaem11@gmail.com

**Keywords:** *SERPING1*, hereditary angioedema, HAE, C1-INH, splicing, intron

## Abstract

Variants that affect splice sites comprise 14.3% of all pathogenic variants in the *SERPING1* gene; more than half of them are located outside the canonical sites. To make a clinical decision concerning patients with such variants, it is essential to know the exact way in which the effect of the variant would be realized. The optimal approach to determine the consequences is considered to be mRNA analysis. In the current study, we present the results of functional analysis of two previously non-described variants in the *SERPING1* gene (NM_000062.3) affecting intron 4: c.686-1G>A and c.685+4dup, which were detected in members of two Russian families with autosomal dominant inheritance of angioedema type 1. Analysis of the patients’ mRNA (extracted from whole blood) showed that the *SERPING1*(NM_000062.3):c.685+4dup variant leads to the loss of the donor splice site and the activation of the cryptic site in exon 4: r.710_745del (p.Gly217_Pro228del), while the *SERPING1*(NM_000062.3):c.686-1G>A variant leads to the skipping of exon 5: r.746_949del (p.Asp229_Ser296del).

## 1. Introduction

Hereditary angioedema (HAE, OMIM#106100) is a disorder characterized by recurrent attacks of severe subcutaneous or submucosal swelling. The attacks may cause acute pain and even death if the edema is localized around the airway. The frequency of this disorder is one per 50,000–100,000 people; it has no ethnic specificities [1].

HAE is subdivided into three main types, depending on the immunological features. HAE-1 is characterized by a general decrease in C1 inhibitor (C1-INH) levels, determined via antigen methods, and a decrease in the protein function. In the case of HAE-2, the C1 inhibitor function is disrupted, while its concentration is normal. HAE-3 is not connected to C1-INH; it is caused by pathogenic variants in other genes: *F12* [2], *PLG* [3], *ANGPT* [4], *MYOX* [5], *KNG1* [6], and *HS3ST6* [7]. Pathogenic variants in the *SERPING1* gene are the cause of HAE in 86.8% of all affected families [8]. The product of the *SERPING1* gene is the C1-INH protein—a serine protease inhibitor. A defect in the inhibitor function leads to an increase in the production of kininogen and bradykinin, which leads to an increase in capillary permeability, blood vessel dilatation, and spasms in smooth musculature, leading to edema. C1-INH controls the components of the complement system and fibrinolysis [9].

In most HAE cases, the decrease in C1-INH levels to less than 35% of the normal values, which leads to swelling, is caused by haploinsufficiency due to *trans*-regulation as a result of the dominant negative effect of the pathogenic variants, the decrease in the enzyme production or the increase in its catabolism. In the case of HAE-2 and a normal concentration of C1-INH, the disorder is caused by an alteration in protein conformation. Usually, the conformation is affected by pathogenic variants involving the following amino acid residues: p.Ala443-Arg444, p.Gln201, and p.Lys251 [10].

The *SERPING1* gene is composed of eight exons and seven introns. Normally, this gene undergoes alternative splicing; however, the physiological role of alternative transcripts is unclear to this day [11,12]. All types of pathogenic variants are described for the gene: missense—32.2%, minor deletions and insertions—36.2%, deletions and duplications—8.3%, splicing affected variants—10.6%, nonsense—9.1% and promoter—3.7%? The introns of the gene include a large number of Alu-repeats, recombinations between which lead to deletions and duplications of exons; in most cases, exon 4 is affected. Missense replacements usually disrupt the structure of the serpin domain of the protein. 36.2% of the variants are minor deletions and insertions, most of which lead to a frameshift [8].

Variants that affect splice sites comprise 14.3% of all pathogenic variants in the *SERPING1* gene. Half of them are located in conservative regions of the splice sites (±1, ±2) [10]. The pathogenicity of these variants is easily predictable using bioinformatic in silico algorithms. However, pathogenicity predictions for the variants in non-canonical sites, as well as the ones affecting the splicing regulatory elements (SRE)—enhancers and silencers [13]—are significantly less reliable. It is also difficult to predict the exact way the effect of the splicing variant would be realized: skipping of one or multiple exons, intron retention, activation of the cryptic splice site, or a combination of all of the above [14]. Therefore, diagnostic decisions, especially concerning a disorder caused by variants in non-canonical splice sites, should ideally be based on factual data about the variants’ effect on the cDNA level obtained during in vitro experiments [15]. It is worth noting that testing RNA extracted from affected tissue is more informative than minigene system analysis, seeing that the genetic context can influence the RNA maturation process. It has been shown that despite the fact that the minigene system shows the presence/absence of the variant’s effect on splicing quite precisely, the information on the exact way of realization and the products obtained as a result of aberrant splicing may often differ from the results of experiments on RNA extracted from the patients’ tissues [16].

In the current study, we present the results of functional analysis of two previously non-described variants in the *SERPING1* gene (NM_000062.3), which affect intron 4: c.686-1G>A and c.685+4dup, detected by us in two members of families from Vologda, Russia, with autosomal dominant inheritance of angioedema type 1.

## 2. Materials and Methods

DNA was extracted from whole blood samples using a Wizard^®^ Genomic DNA Purification Kit (Promega, Madison, WI, USA) according to the manufacturer’s protocol.

PCR primer sequences were chosen according to the *SERPING1*(NM_000062.3) reference sequence. Quantitative analysis was carried out using the SALSA MLPA Probemix SALSA MLPA Probemix P243 SERPING1-F12 (MRC-Holland, Amsterdam, The Netherlands). The SALSA MLPA Probemix P243-B1 *SERPING1-F12* contains 33 MLPA probes with amplification products between 168 and 500 nucleotides (nt). This includes eight probes targeting all exons of the *SERPING1* gene and 13 probes for the F12 gene, targeting all exons with the exception of exon 3. In addition, 11 reference probes that detect autosomal chromosomal locations are included.

Total RNA was extracted from whole blood samples using ExtractRNA BC032 (Evrogen, Moscow, Russia). cDNA from the RNA matrix was obtained using an MMLV RT Kit SK021 (Evrogen, Moscow, Russia). Sequence fragments were amplified using the patient’s and relatives’ cDNA as a template, with the following primers: C1JUNCT3-4F: CAGGTCCTGCTCGGGGCTG and C1JUNCT6-7R: GCCCCACCTTGGCTTTCAAAG (Figure 1).

The PCR products were examined using the amplification fragment length polymorphism analysis. After PCR amplification and polyacrylamide gel electrophoresis (7% and 9%) with ethidium bromide staining, the obtained products were visualized in UV light.

Automated Sanger sequencing was carried out using an ABIPrism 3500xl Genetic Analyzer (Applied Biosystems, Foster City, CA, USA) according to the manufacturer’s protocol.

This study was conducted according to the guidelines of the Declaration of Helsinki and approved by the Institutional Review Board of the Research Center for Medical Genetics, Moscow, Russia (approval number 2018-3/4). The probands gave informed consent to the genetic testing and the publication of anonymized data.

## 3. Results

We examined members of two families with a diagnosis of HAE-1. The pedigrees are presented in Figure 1. Both families had autosomal dominant inheritance of the disorder (Figure 2).

In the HAE_728 family, the inheritance of the disease was traced in 10 patients from four generations; seven patients aged from 21 to 59 years were available for examination. The proband in family 728—III.3 (Figure 2a, marked with an arrow)—was a 59-year-old woman. She initially consulted an immunologist in March 2014 with complaints of weekly pale dense edema on her face, neck, feet, hands, thighs, and genitals, as well as laryngeal edema 2–4 times per year. The disease manifested at the age of 2–3 years as recurrent abdominal pain (without an apparent cause), nausea, and vomiting; therapy did not have any effect. The proband was monitored by a gastroenterologist with various diagnoses. When she reached puberty (12 years), the peripheral edema of varying localization (face, limbs, body, genitals) manifested. The symptoms lasted from 5 to 7 days; systemic glucocorticosteroids and antihistamine drugs did not have any effect. The edema could be triggered by injury, physical exercise, stress, menstruation, or sometimes occurred without an apparent cause. The patient also complained of periodical intense abdominal pain with vomiting or diarrhea. She was hospitalized multiple times with suspected acute surgical pathology, ultrasound scanning detected fluid in the abdominal cavity, and diagnostic laparoscopy was carried out twice. At the moment of examination, the C1 inhibitor level was 4.3 mg/dL (reference values 15–35 mg/dL), functional C1–17.5% (reference values 70–130%). Sequencing of the *SERPING1* gene for patients 728 III.2, 728 III.3, 728 III.5, 728 IV.1, 728 IV.2, 728 IV.3, and 728 IV.4 detected the *SERPING1*(NM_000062.3):c.685+4dup variant in a heterozygous state in intron 4 (Figure 2b). The unaffected family members (II.4 and III.7) had no nucleotide sequence alterations. This variant was absent in the gnomAD database (as of 16.09.2023) [17]. According to the SpliceAI [18] algorithm, insertion of a nucleotide in the donor splice site region can lead to the loss of the donor splice site with a Δ score of 0.03 and the gain of a new donor splice site in the position of −40 from the *SERPING1*(NM_000062.3):c.650G with a Δ score of 0.19. Both Δ score values are lower than the recommended cut-off point of 0.2 and much lower than the recommended significance point of 0.5. The effect of this variant on splicing is also predicted by the programs MMSplice [19] and NetGene2 [20].

In the HAE_1531 family (Figure 2b), the disease inheritance was traced in four patients from two generations; the patients were aged from 30 to 52 years, and three of them were available for examination. The proband in family 1531—II.1 (Figure 2c, marked with an arrow)—was a 52-year-old woman who initially consulted an immunologist in March 2022. The disease manifested at the age of approximately 20 years as pain in the upper abdomen, inflation, vomiting with bile, severe weakness, and diarrhea. The pain could be triggered by menstruation, sometimes did not have an apparent cause, lasted up to 2–3 days, and stopped independently of the conducted therapy. The proband notes peripheral pale, thick, non-itchy hand edema since the age of 40 years and facial edema since the age of 45 years. The edema could be triggered by mechanical stimuli or sometimes had no apparent cause; systemic glucocorticosteroids and antihistamine drugs had no effect. The C1 inhibitor level was 3.7 mg/dL (reference values 15–35 mg/dL), functional C1–27.2% (reference values 70–130%). Sequencing detected the *SERPING1*(NM_000062.3):c.686-1G>A variant in a heterozygous state in intron 4 (Figure 2d). Aside from the patients, two asymptomatic 3-year-old family members (1531 IV.1, 1531 IV.2) were also examined at the request of their legal representatives and had the c.686-1G>A variant as well. According to the SpliceAI algorithm [18], this variant may lead to the loss of the acceptor splice site with a Δ score of 0.97 and the gain of a new donor site in the position of +14 from the replacement with a Δ score of 0.60. This variant was absent from the gnomAD database (as of 16.09.2023) [17]. The effect of this variant on splicing is also predicted by the programs SPiP: 98.67% [96.17–99.55%] [21], NetGene2 [20], and MMSplice [19]. Another nucleotide sequence variant, c.686-1G>T, was described in the same position twice [12,22]; however, no functional analysis of its effect has been carried out.

The effects of these variants were analyzed on mRNA level using RT-PCR (Figure 3). The mRNA was extracted from the available patients’ blood (728 III.3; 728 III.5; 1531 II.1; 1531 III.1; 1531 III.4).

The patients from the HAE_1531 family with the *SERPING1*(NM_000062.3):c.686-1G>A variant had a more intense PCR product with a shorter fragment length than expected for the canonical transcript (Figure 2a). Sanger sequencing for this product showed that it corresponds to the skipping of exon 5: r.746_949del (Figure 2d). The loss of 204 nucleotides leads to the deletion of 68 amino acids: p.Asp229_Ser296del, as predicted by the SpliceAI program.

Sanger sequencing of the cDNA amplification product for the patients from the HAE_728 family did not detect any difference from the reference transcript. However, electrophoresis showed heteroduplexes that distinguished them from the control samples. Seeing that heteroduplexes are a product of conformational interaction between two chains with different nucleotide compositions, we decided to examine the PCR product, enriching it with the altered allele. The product was visualized during the analysis on polyacrylamide gel with higher resolution; its expression level was significantly lower in comparison to the normal allele, preventing its detection via Sanger sequencing. However, the analysis of the PCR product enriched with the altered allele, obtained as a result of heteroduplex amplification, allowed us to establish that the *SERPING1*(NM_000062.3):c.685+4dup variant in the cDNA sequence leads to the loss of the donor splice site and the activation of the cryptic site in exon 4: r.710_745del, thus leading to the loss of 12 amino acids in the serpin domain of the protein: p.Gly217_Pro228del.

## 4. Discussion

The conducted examination allowed us to determine the effect of two previously non-described variants in intron 4 of the *SERPING1*(NM_000062.3) gene—c.685+4dup and c.686-1G>A—detected during molecular genetic analysis in two Russian families with dominant hereditary angioedema type 1.

It was shown that the *SERPING1*(NM_000062.3):c.685+4dup variant leads to the loss of the donor splice site and the activation of the cryptic site in exon 4: r.710_745del (p.Gly217_Pro228del), as predicted by the SpliceAI algorithm (with very low scores). This alteration does not lead to the formation of a premature termination codon; with that, the aberrant to normal transcript ratio was quite low, which could not explain the substantial functional inhibitor decrease (17.5%) in the proband. Seeing that the transcript with r.710_745del is not a normal product of alternative *SERPING1* splicing, one could speculate that most of it degrades by the No-go decay (NGD) mechanism [23], which was also suggested for other variants affecting splice sites of the *SERPING1* gene [8]. Other missense variants that cause hereditary angioedema have been described in this region of the gene: c.653T>A (p. Val218Asp) [24], c.671T>G (p. Ile224Ser) [25], c.671T>A (p. Ile224Asn) [26], c.674T>C (p. Phe225Ser) [26].

The *SERPING1*(NM_000062.3):c.686-1G>A variant leads to the skipping of exon 5: r.746_949del (p.Asp229_Ser296del). This deletion disrupts the structure of the C1-INH serpin domain. The same transcript is present in the control samples and described as the result of alternative splicing of the *SERPING1* gene—a protein-encoding transcript ENSG00000149131. However, the levels of this transcript are normally extremely low, while the c.686-1G>A variant leads to all alleles that carry it being spliced via the formation of a minor transcript; therefore, the amount of the normal transcript from the other allele is not enough for the C1-INH inhibitor to function. The proband from the HAE_1531 family has a decrease in functional C1 inhibitor levels to 27.2%, which could not be explained solely by the skipping of exon 5 in half of the transcripts. Seeing that the ENSG00000149131 transcript is normally present in all healthy people, its product is unlikely to cause a dominant negative effect on the normal C1-INH protein. On the other hand, during RNA analysis, we did not detect the transcript predicted to be a result of the formation of a new donor splice site in the position of +14 from the replacement. The length of this transcript is not a multiple of three nucleotides and should lead to the formation of a premature termination codon due to a frameshift. Thus, its absence may be a result of nonsense-mediated RNA decay and be the second mechanism causing haploinsufficiency, which leads to the development of angioedema in patients with the c.686-1G>A variant. Exon 5’s missense variants of the *SERPING1* gene have been repeatedly described as the cause of hereditary angioedema. HGMD contains more than 30 different amino acid substitutions in this area [27].

Twelve other variants located in intron 4 of the *SERPING1* gene were described in patients with hereditary angioedema in scientific publications and the HGMD database (Table 1). We analyzed the effects predicted by an in silico SpliceAI program, as well as the results of in vitro experiments studying the effects these variants have on splicing.

It is worth noting that exons 4 and 5 adjacent to intron 4 of the *SERPING1* gene have a number of nucleotides that are a multiple of 3; therefore, in this case, skipping leads to the formation of a product that does not undergo NMD.

As shown in Table 1, both in silico and in vitro systems are able to successfully predict the effect of the variants on splicing; however, the exact effect of each particular variant could be seen only during RNA analysis. Not all predictions are realized in the RNA structure; aside from that, minigene system analysis provides a greater number of variable transcripts than present in the patient’s RNA or predicted in silico, which shows the important role genetic context plays in the realization of regulatory region variant effects. The further the variant is located from the canonical splice site, the higher the probability of its effect being realized via cryptic splice site activation rather than via exon skipping. Variants affecting nucleotides ±1 and ±2 lead to the deletion of exons with disrupted donor or acceptor splice sites; at the same time, variants affecting nucleotides ±5 or further lead to the formation of products with altered exon length.

Some products predicted in silico cannot be detected during either minigene analysis or RNA analysis, which is caused by the frameshift activating the NMD mechanism. The possibility of the formation of such products, in addition to the NGD effect, may explain the decrease in the mutant allele expression.

## 5. Conclusions

The conducted study allowed us to confirm the pathogenicity of two previously non-described variants in intron 4 of the *SERPING1* gene. Establishing the variant pathogenicity in one of the families helped determine the status of three asymptomatic children, which is essential for the prevention of life-threatening complications of the disease.

We have established the effect of two novel variants affecting splicing sites in intron 4 of the *SERPING1* gene.

## Figures and Tables

**Figure 1 biomedicines-12-00072-f001:**
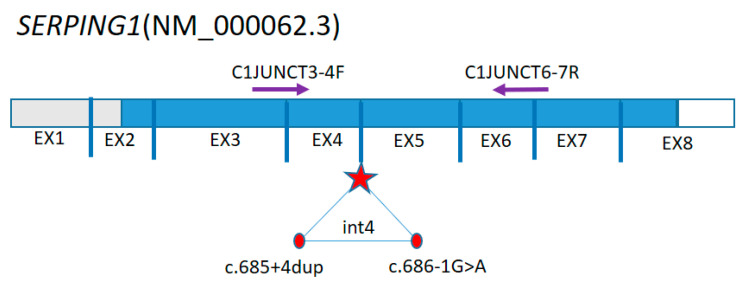
The scheme of primers arrangement on the SERPING1(NM_000062.3) sequence. The sequences encoding proteins are colored blue. An asterisk marks the junction of exons 4 and 5. The variants c.686-1G>A and c.685+4dup are located in this intron 4.

**Figure 2 biomedicines-12-00072-f002:**
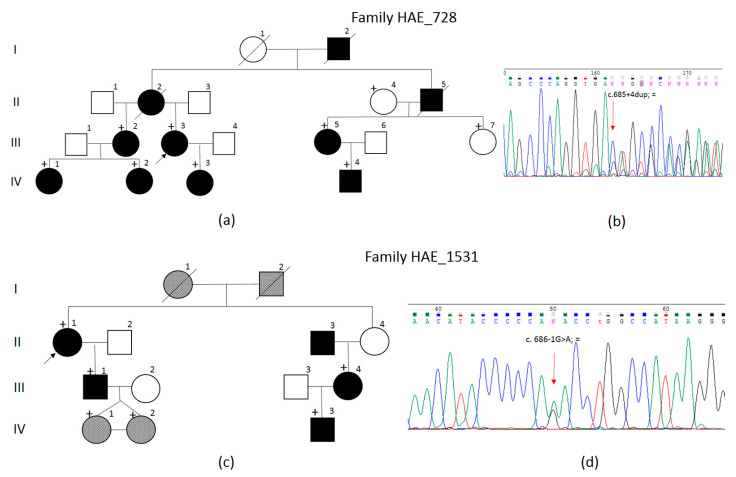
Pedigrees of the HAE_728 (**a**) and HAE_1531 (**c**) families and results of direct automated Sanger sequencing of the patients’ genomic DNA for HAE_728 (**b**) and HAE_1531 (**d**). The black arrows indicate the probands. Black shapes indicate affected family members, white—healthy, gray—non-examined, or members with unknown status. Plus signs indicate the patients whose blood was obtained for DNA analysis. Red arrows indicate the *SERPING1* (NM_000062.3):c.685+4dup and *SERPING1* (NM_000062.3):c.686-1G>A variants.

**Figure 3 biomedicines-12-00072-f003:**
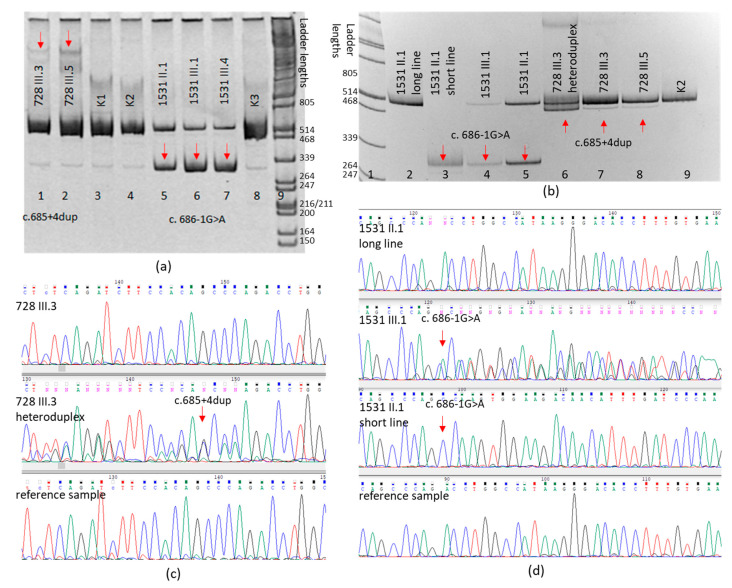
PCR analysis results for the cDNA of patients with the *SERPING1*(NM_000062.3):c.685+4dup and *SERPING1*(NM_000062.3):c.686-1G>A variants. Electrophoresis results (**a**,**b**) and Sanger sequencing results for cDNA of the patient with the c.685+4dup variant (**c**) and the patient with the c.686-1G>A variant (**d**). (**a**) Lanes 1 and 2—cDNA amplification products for patients from the HAE_728 family, arrows indicate heteroduplexes; lanes 3, 4, 8—PCR products for the control cDNA samples; lanes 5, 6, 7—cDNA PCR products for patients from the HAE_1531 family; lane 9—molecular weight marker (λ phage DNA digested by restriction endonuclease PstI); (**b**) lane 1—molecular weight marker (λ phage DNA digested by restriction endonuclease PstI), lane 2—PCR product for the long fragment of the 1531 II.1 sample, lane 3—PCR product for the short fragment of the 1531 II.1 sample; lanes 4, 5—PCR products for both fragments of the 1531 III.1 and 1531 II.1 samples; lane 4—PCR product for the 728 III.3 heteroduplex; lanes 7, 8—cDNA PCR products for probands from the HAE_728 family; lane 9—PCR product for the control cDNA sample.

**Table 1 biomedicines-12-00072-t001:** Splice site variants in intron 4 of the *SERPING1* gene detected in patients with HAE.

cDNA Variant	Functional Proof	In Vitro Effect	In Silico Effect (SpliceAI (ΔScore))	References
c.685+1G>T	no		Skipping of exon 4 (1); Shortening of exon 4 by 36 bp (0.71)	[28]
c.685+1G>A	no		Skipping of exon 4 (1); Shortening of exon 4 by 36 bp (0.68)	[26]
c.685+1del	MGA *, BDR *	MGA: skipping of exon 4, activation of cryptic intronic sites. BDR: skipping of exon 4 and exons 4–5	Skipping of exon 4 (1); Shortening of exon 4 by 1 bp (NMD) (0.68)	[16]
c.685+2T>G	no		Skipping of exon 4 (0.99); Shortening of exon 4 by 36 bp (0.74)	[28]
c.685+2T>A	BDR	Skipping of exon 4. Decrease in the expression of the aberrant transcript due to NGD *	Skipping of exon 4 (1); Shortening of exon 4 by 36 bp (0.62)	[29]
c.685+2_685+13del	MGA, BDR	MGA: skipping of exon 4, activation of cryptic intronic sites. BDR: skipping of exon 4 and exons 4–6	Skipping of exon 4 (1); Shortening of exon 4 by 36 bp (0.56)	[16]
c.685+4dup	BDR	BDR: exon 4 shortened by 36 bp. Decrease in the expression of the aberrant transcript due to NGD	Skipping of exon 4 (0.03); Shortening of exon 4 by 36 bp (0.19)	[this study]
c.685+5G>A	no		Skipping of exon 4 (0.04); Shortening of exon 4 by 36 bp (0.20)	[10]
c.685+5G>T	no		Skipping of exon 4 (0.04); Shortening of exon 4 by 36 bp (0.20)	[10]
c.685+31G>A	no		No effect	[6]
c.686-7C>G	BDR	Activation of the cryptic intronic site, inclusion of 2 amino acid residues (+6 nucleotides)	Skipping of exon 5 (0.92); Elongation of exon 5 by 6 bp (0.99)	[12,25]
c.686-12A>G	MGA	Activation of the intronic cryptic site, elongation by 11 nucleotides, NMD *	Skipping of exon 5 (0.71); Elongation of exon 5 by 11 bp (0.94)	[16]
c.686-1G>A	BDR	Skipping of exon 5	Skipping of exon 5 (0.97); Shortening of exon 5 by 1 bp (0.60)	[this study]
c.686-1G>T	no		Skipping of exon 5 (0.96); Shortening of exon 5 by 1 bp (0.66)	[22]

* MGA—Minigene assay, BDR—blood-derived mRNA, NGD—No-go decay, NMD—nonsense-mediated decay.

## Data Availability

We can provide them or conduct a reanalysis upon request by mail schagina@med-gen.ru. Raw data cannot be posted due to patient confidentiality. All sequences and electrophoregrams related to the experiments of this work are presented in the figures in the manuscript.

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
