# Peer review of "Functional Characterization of Two Novel Intron 4 SERPING1 Gene Splice Site Pathogenic Variants in Families with Hereditary Angioedema"

_biomedicines, 2023, doi:10.3390/biomedicines12010072_

Round 1

Reviewer 1 Report

Comments and Suggestions for Authors

This is a well-written and technically well-substantiated and structured article. Two splicing variants identified in two families are reported. One of these variants in the canonical splicing acceptor site and the other two base pairs away from the canonical splicing donor site have an effect on splicing that is adequately demonstrated through mRNA analysis and Sanger sequencing.

The contribution of the reported findings is discrete since the position of the variants was expected to have an effect on splicing and the software used to predict the effect of the variants already indicated this possible effect with high susceptibility.

It seems to me that the main contribution of this work should focus on the evaluation of other software for predicting the effect of splicing variants and the relevance of performing the analysis of the effect on the patient's mRNA. Therefore, I recommend the use of other bioinformatics tools that allow comparison of their predictions with the effects observed in these variants.

The discussion should place greater emphasis on establishing whether the deleterious effect of these variants is due to the functional effect on the protein of the deletions of amino acid residues resulting from aberrant splicing or whether it is mainly due to a phenomenon of mRNA decay. For this I would recommend a review of the variants reported in the regions of the protein that undergo deletion and their classification, as well as a review of the splicing variants and whether there are reports on whether they undergo RNA decay.

Conclusions should be focused on the relevance of mRNA analysis in patients with splicing variants mainly when they are variants not previously reported.

Author Response

We are grateful for your attentive review. The answers to your comments and comments are given below.

The contribution of the reported findings is discrete since the position of the variants was expected to have an effect on splicing and the software used to predict the effect of the variants already indicated this possible effect with high susceptibility.

Thank you for your appreciation of the technical implementation of our idea. The appearance of many works currently shows the importance not only to assess the effect of substitution on splicing as a fact, but also to determine the specific effect that this option leads to. This knowledge is important for predicting the severity of changes at the protein level and, ultimately, predicting the severity of the phenotypic effects of the nucleotide sequence variant.

It seems to me that the main contribution of this work should focus on the evaluation of other software for predicting the effect of splicing variants and the relevance of performing the analysis of the effect on the patient's mRNA. Therefore, I recommend the use of other bioinformatics tools that allow comparison of their predictions with the effects observed in these variants.

We have analyzed the options using other bioinformatic algorithms. Information about this was added by us in the manuscript (highlighted in yellow).

The discussion should place greater emphasis on establishing whether the deleterious effect of these variants is due to the functional effect on the protein of the deletions of amino acid residues resulting from aberrant splicing or whether it is mainly due to a phenomenon of mRNA decay. For this I would recommend a review of the variants reported in the regions of the protein that undergo deletion and their classification, as well as a review of the splicing variants and whether there are reports on whether they undergo RNA decay.

Thanks for the comment. None of the variants of the nucleotide sequence presented by us leads to the breakdown of the protein. We focused on analyzing the spectrum of missense variants described in the regions of emerging deletions. The information has been added to the manuscript, highlighted in yellow.

Conclusions should be focused on the relevance of mRNA analysis in patients with splicing variants mainly when they are variants not previously reported.

Thank you. A relevant phrase has been added to the conclusions.

Reviewer 2 Report

Comments and Suggestions for Authors

The authors presented the functional analysis of two previously non-described variants of SERPING1 gene in intron 4: c.686-1G>A and c.685+4dup, which was associated with angioedema type 1.

1. In general, overall data is insufficient.

2. Illustrate the position of c.686-1G>A and c.685+4dup and the location of the designed primers in gene transcript.

3. Whether the mutation of c.686-1G>A and c.685+4dup affect the expression of SERPING1. qPCR is required for SERPING1 gene quantification and expression of variable splicing transcripts after mutation.

4. Add the Sanger sequence of healthy control in Figure 1.  

5. Mark the length of PCR products in Figure 2a and 2b.

6. The authors only studied the main transcript of SERPING1 (NM_000062.3), whether other transcripts were also affected, or whether other transcripts would also affect the detection of this transcript.

Comments on the Quality of English Language

Minor editing.

Author Response

We are grateful for your attentive review. The answers to your comments and comments are given below. Your comments have helped us to significantly improve the presentation of our work.

  1. In general, overall data is insufficient.

The manuscript has been edited in accordance with the comments of the reviewers

  1. Illustrate the position of c.686-1 G>A and c.685+4 dup and the location of the designed primers in gene transcript.

The primer sequences are given in the materials and methods section. We have added a figure 1. We indicated the location of the primers and the localization of the variants.

  1. Whether the mutation of c.686-1G>A and c.685+4dup affect the expression of SERPING1. qPCR is required for SERPING1 gene quantification and expression of variable splicing transcripts after mutation

In a direct experiment with PCR and foresis, it can be seen that the SERPING1(NM_000062.3):c.685+4 dup variant leads not only to the formation of a shortened product. This product is significantly less than expected if this option would not have led to the formation of another product undergoing NMD. Undoubtedly, this variant affects expression and it has been established. This fact is discussed in the text of the manuscript. However, in this situation, we cannot come up with a clinical or scientific reasons for the need to confirm this fact by qPCR.

  1. Add the Sanger sequence of healthy control in Figure 1.

Thanks for the comment. The figure is completed.

  1. Mark the length of PCR products in Figure 2a and 2b.

The lengths of the allele ladderer are added to the figure

  1. The authors only studied the main transcript of SERPING1 (NM_000062.3), whether other transcripts were also affected, or whether other transcripts would also affect the detection of this transcript.

We studied the reference transcript. Alternative splicing has been described for the SERPING1 gene. However, when examining nucleic acids isolated directly from patients' blood cells, we did not see significant amounts of alternative transcripts, with the exception of the transcript without exon 5. Since all transcripts of the gene have the same boundaries, we assume that these splicing variants will lead to the same changes that we observe in the main transcript.

Reviewer 3 Report

Comments and Suggestions for Authors

Introduction:

LINE 28: Hereditary angioedema (HAE, OMIM#106100)) is....

Line 54-58: describe the % of diferents variants: missense 32,3%,frameshift,  splicing,

the SERPING1 gene is the only pathogenic gene known to be related to HAE in the Russian population?  The approach of paper is only use sanger sequencing to detect variants in SERPING1.

Materials and Methods:

Line 84: MLPA is not a quantitative technique is semiquantitative. Explain how many probes of SERPING1 include P243

Results:

Legend of Figure 1 and 2 are too long, especially figure 2 and specifically the description of lane 9.

Discussion

Comment that c.686-1G>A is described in clinvar and LOVD and c.685+4dup not described in clinvar and LOVD

Author Response

We are grateful for your attentive review. Your comments have helped us to significantly improve the presentation of our work.

LINE 28: Hereditary angioedema (HAE, OMIM#106100)) is....

Тhe correction was made

Line 54-58: describe the % of diferents variants: missense 32,3%,frameshift,  splicing,

 Information added to the manuscript

the SERPING1 gene is the only pathogenic gene known to be related to HAE in the Russian population?  The approach of paper is only use sanger sequencing to detect variants in SERPING1.

Of course, SERPING1 is not the only gene responsible for HAE in Russian patients. However, as in all countries, this gene is a major one and its study by Sanger sequencing is justified by both the frequency and the small size of the gene.

Materials and Methods:

Line 84: MLPA is not a quantitative technique is semiquantitative. Explain how many probes of SERPING1 include P243

 Information added to the manuscript

Results:

Legend of Figure 1 and 2 are too long, especially figure 2 and specifically the description of lane 9.

We tried to break the drawings into parts, but it turned out that then the meaning of the illustrations was lost.

Discussion

Comment that c.686-1G>A is described in clinvar and LOVD and c.685+4dup not described in clinvar and LOVD

Thanks for the question. We checked again.

Reviewer 4 Report

Comments and Suggestions for Authors

The author confirmed the findings of a functional investigation of two previously unknown variations in the SERPING1 gene (NM_000062.3) that affect intron 4: c.686-77 1G>A and c.685+4dup, which we discovered in two members of families with autosomal dominant angioedema type 1. The in silico analysis was verified by mRNA analysis. The results reported is also corroborated by an extensive examination of the family, which confirms what has been observed experimentally. Only the correct nomenclature of the variants in lane 50 should be indicated, and the term mutation should be changed to pathogenic variants in the title and text.

Author Response

We would like to thank you for your attentive review and high appreciation of our research. The terms have been replaced.

Round 2

Reviewer 1 Report

Comments and Suggestions for Authors

The authors have addressed all the previously mentioned issues and have made all the necessary changes to the manuscript.

Author Response

We sincerely thank the Reviewer for his attentive and friendly attitude to our manuscript.

Reviewer 2 Report

Comments and Suggestions for Authors

1. References are missing.

2. Where are the positions of c.686-1 G>A and c.685+4 dup.

Comments on the Quality of English Language

Minor editing.

Author Response

We sincerely thank the Reviewer for his attentiveness and thoroughness. Your comments have helped to significantly improve our manuscript.

  1. References are missing.

Thanks! References have been added.

2. Where are the positions of c.686-1 G>A and c.685+4 dup.

Changes have been made to Figure 1.